

# Hybrid insolation forcing of Pliocene monsoon dynamics in West Africa

Rony R. Kuechler, Lydie M. Dupont*, Enno Schefuß

MARUM – Center for Marine Environmental Sciences, University of Bremen, Leobener Str. 8, 28359 Bremen, Germany

*Correspondence to*: Lydie Dupont (ldupont@marum.de)

**Abstract.** The Pliocene is regarded as a potential analogue for future climate with conditions generally warmer-than-today and higher-than-preindustrial atmospheric $CO_2$ levels. Here we present the first orbitally resolved records of continental hydrology and vegetation changes from West Africa for two Pliocene time intervals (5.0-4.6 Ma, 3.6-3.0 Ma). Our results indicate that changes in local insolation alone are insufficient to explain the full degree of hydrologic variations. Generally

two modes of interacting insolation forcings are observed: during eccentricity maxima, when precession was strong, the West African monsoon was driven by summer insolation; during eccentricity minima, when precession-driven variations in local insolation were minimal, obliquity-driven changes in the summer latitudinal insolation gradient became dominant. This hybrid monsoonal forcing concept explains orbitally-controlled tropical climate changes, incorporating the forcing mechanism of latitudinal gradients for the Pliocene, which probably increased in importance during subsequent Northern

Hemisphere glaciations.

## 1 Introduction

The hydrologic cycle is of vital importance for the global climate system, owing to its function in regulating the heat and moisture balance (Lindzen, 1990, 1994). This is mainly achieved through the atmospheric and ocean circulation, which tend to equalize the differences in solar heating between low and high latitudes. The importance of this heat redistribution is

illustrated by the fact that around 50% of the annual energy budget of high latitudes originates from lower latitudes (Peixoto and Oort, 1992), peaking in winter, when polar night conditions prevail at high latitudes (Davis and Brewer, 2011). In this way, atmospheric water vapour acts as the dominant greenhouse gas, exerting an important feedback on global warming (e.g., Held and Soden, 2000). Thus a good understanding of the dynamics of the hydrological cycle is crucial. However, focusing on the instrumental record only would limit the state of our knowledge (Brohan et al., 2006). To assess the

dynamics of a changing climate, paleoclimate studies provide means to investigate long-term developments of the monsoon (Mohtadi et al., 2016).

Monsoon systems are the most prominent parts of the global hydrological cycle. Monsoon variability on orbital time scales is mainly attributed to variations in low-latitude summer insolation, which is dominated by 19 kyr and 23 kyr periodicities of the precession cycle (e.g. Kutzbach, 1981; Rossignol-Strick, 1983; Tiedemann et al., 1994; Larrasoaña et al., 2013; Mohtadi





et al., 2016). Consequently, maximum insolation intensities are expected to increase monsoon circulation and thus rainfall, comparable to modern seasonal variations. Recent sensitivity studies show that the effect of precession on tropical climate is far stronger than that of obliquity (Bosmans et al. 2015a). Nevertheless, also obliquity (41 kyr) cycles have been detected in sequences of Mediterranean sapropels (Lourens et al., 1996), which formation depends on monsoonal fresh-water discharge

from Africa (Rossignol-Strick, 1973; Colleoni et al., 2012; Larrasoaña et al., 2013). Equally, dust flux rates at Ocean Drilling Program (ODP) Site 659 (Tiedemann et al., 1994) and pollen flux rates at ODP Site 658 (Dupont et al., 1989) show obliquity cycles in records of tropical Africa, which are difficult to explain, since obliquity has a negligible effect on low-latitude insolation (Laskar, 1990).

A plausible mechanism that incorporates the obliquity signal and the climatic linkage between low and high latitudes is
found in the latitudinal insolation gradient (LIG; Davis and Brewer, 2009). This intra-hemispheric insolation gradient leads to differential heating between the cold polar regions and the hot tropics and creates the latitudinal temperature/pressure gradient, which ultimately controls the poleward heat and moisture transport, a mechanism that has been invoked in the so-called "gradient hypothesis" by Raymo and Nisancioglu (2003) to explain the strong obliquity rhythm of glacial-interglacial cycles between 0.8-3.0 Ma (the "41 kyr world"). Another gradient hypothesis is favoured by Bosmans et al. (2015b), who
suggest that the inter-hemispheric insolation gradient drives winds and associated cross-equatorial moisture transport deep into the African continent linked with an intensification of the Hadley cell during winter (Reichart, 1997). The same mechanism has been invoked by Leuschner and Sirocko (2003) defining the Indian Summer Monsoon Index. The concept of insolation gradients is supported by recent climate model studies (Mantsis et al., 2014) simulating an enhanced mid-latitude eddy circulation (which is important for the heat and moisture transport from low to high latitudes) in response to low
obliquity (resulting in a strong LIG during summer), and a concomitant shift of the poleward boundaries of the tropical rain belt towards the equator. In addition, weakening of the inter-hemispheric gradient leads to diminished cross-equatorial heat transport and a weaker Hadley circulation during winter (Reichart, 1997; Bosmans et al., 2015a). Further model simulations corroborate the importance of an obliquity-induced insolation gradient forcing for West African Monsoon variability (Rachmayani et al., 2016).

Both gradient hypotheses stay in contrast to the traditional view, in which the obliquity signal in the tropics is linked to changes in the extent of glaciations. Northern Hemisphere ice sheets and sea-ice would affect the monsoon through the impact on the atmospheric circulation and moisture advection (Tiedemann et al., 1994; DeMenocal, 1995; Mohtadi et al., 2016). Using Pliocene records (between 5 and ~3 Ma) that register paleoclimate prior to the intensification of the Northern Hemisphere Glaciation, we can test if the monsoon system responds to obliquity independently of Northern Hemisphere ice
growth and decay.

Another good reason to study the warmer-than-today climate of the Pliocene is that many boundary conditions are similar to today (Dowsett et al., 2013; Haywood et al., 2013). This last warm period of the geological record is often referred to as future analog. For the assessment of Pliocene climate conditions, focus has been on sea surface (Dowsett et al., 2013) and continental temperatures (Salzmann et al., 2013). Global mean annual temperatures were more than 3°C warmer and sea



level was about 22±10 m higher than at present (Dowsett et al., 2013; Haywood et al., 2013). However, data-model mismatches have been detected for terrestrial temperature estimates in the tropics likely due to limited proxy data (Salzmann et al., 2013). Moreover, a model inter-comparison revealed inconsistencies in tropical precipitation estimates (Haywood et al., 2013) indicating that more data on tropical hydrology are needed.

Hydrogen isotopes (Deuterium) of plant leaf waxes ($\delta D_{wax}$) provide a proxy for paleohydrologic variations (Fig. 1) related to the isotopic composition of precipitation ($\delta D_{rain}$; Sachse et al., 2012). In tropical regions, $\delta D_{rain}$ is negatively correlated to precipitation amounts (Dansgaard, 1964; Rozanski et al., 1993; Risi et al., 2008), whereas $\delta D_{wax}$ also incorporates secondary effects of evapotranspiration, plant physiology, and photosynthetic pathway (Sachse et al., 2012). In areas with C4 grasses, the stable carbon isotopic compositions of the same compounds ($\delta^{13}C_{wax}$) provide additional information about vegetation

changes and can be used to differentiate between contributions from grassy (C4) and woody (C3) vegetation (e.g., Vogts et al., 2009).

Our study focusses on ODP Site 659, situated offshore of West Africa and centered beneath the main wind trajectories (Fig. 2). Its dust record provides evidence for the persistence of orbital arid-humid cycles during the last 5 Ma (Tiedemann et al., 1994). Coupled $\delta D_{wax}$ and $\delta^{13}C_{wax}$ analyses from this site for the Last Glacial Cycle have demonstrated their ability to record

shifts in West African hydrology and vegetation (Kuechler et al., 2013). However, $\delta D_{wax}$ studies from West Africa covering the Pliocene have not yet been published. Here we provide orbitally resolved $\delta D_{wax}$ and $\delta^{13}C_{wax}$ records for two time intervals (5.0-4.6 Ma and 3.6-3.0 Ma) to evaluate hydrologic changes in Pliocene West Africa. In addition to the well-established precession forcing, we discuss further insolation mechanisms to explain the long-term evolution of the West African hydrologic cycle.

**1.1 Modern climate and vegetation**

The amount and distribution of precipitation in West Africa is a function of latitude (Nicholson, 2009; Fig. 3). Tropical rainfall is tightly linked to atmospheric dynamics at higher altitudes, involving the African Easterly Jet (AEJ) and the Tropical Easterly Jet (Nicholson, 2009). Highest precipitation amounts are found in the equatorial regions, which are marked by a double rainfall maximum, occurring during the transition seasons, which is mostly attributed to the latitudinal migration

of the ITCZ. Further north a single rainfall maximum occurs during summer (July/August), when a low pressure system is formed over the Sahara due to stronger heating of the northern African continent relative to the adjacent ocean. This low pressure system is situated between the NE trade winds and the moist "SW monsoon". Precipitation during the wet season in Sahara and Sahel moderates the surface temperatures, thus enhancing the temperature contrast to the hyperarid Sahara in the north, which strengthens the AEJ during its northward displacement after rainfall (Nicholson and Grist, 2003). Precipitation

in the temperate regions of North Africa is influenced by the mid-latitude westerlies falling mainly during the winter season. The Sahara desert separates the summer and winter precipitation regimes and thus, exhibits a complex pattern. Much of the rainfall in the desert occurs during the transition seasons and has an extra-tropical origin related to the mid-latitude westerlies interacting with the tropical easterlies (Nicholson, 1981, 2000).



The modern vegetation distribution and composition in West Africa (White, 1983) reflects the latitudinal migration of the tropical rain belt (Fig. 3). Close to the equator, high precipitation rates lead to dense vegetation cover, consisting of tropical rain forests and woodlands (mostly C3). Further to the north, towards the Sahara desert, increasingly open grasslands (mostly C4) occur as rainfall decreases and wet season length shortens. North of the Sahara, the Mediterranean vegetation is

composed of scrublands, steppes and forests, which consist exclusively of C3 plants.

## 2 Material & Methods

ODP Site 659 is located on top of the submarine Cape Verde Plateau at 3070 m water depth (Fig. 2; Ruddiman et al. 1987). The siliciclastic fraction is considered to be of purely eolian origin due to low carbonate concentrations in the dust composition and its distal location on a submarine rise excluding fluvial input (Tiedemann et al., 1994). Two Pliocene age

models have been established, one based on variations in dust accumulation (Tiedemann et al., 1994) and the other based on stable oxygen isotopes ($\delta^{18}$O) of benthic foraminifers (Clemens, 1999). The latter age model is used in this study, since it is independent from the dust record and produces a better fit with the global benthic $\delta^{18}$O stack (Lisiecki and Raymo, 2005), especially for the Zanclean.

Stable isotope analyses of hydrogen ($\delta$D) and carbon ($\delta^{13}$C) were carried out on 230 samples with an average temporal

resolution of ~4 kyr, comparable to the dust record of the same site (Tiedemann, 1991; Tiedemann et al., 1994). A sampling gap between 3.33-3.29 Ma is related to a core break. A detailed description of methods, including lipid extraction, quantification and stable isotope analyses, is given in Kuechler et al. (2013). $\delta$D values are reported relative to Vienna Standard Mean Ocean Water (VSMOW). An external $n$-alkane standard yielded a precision ($1\sigma$) and accuracy of 2‰, and squalane as internal standard yielded values of 2‰ and 1‰, respectively. The mean precision ($1\sigma$) of replicates for the

$n$-C$_{29-33}$ alkanes is 2‰. $\delta^{13}$C values are reported relative to the Vienna Pee Dee Belemnite (VPDB) standard. The external standard yielded a precision ($1\sigma$) of 0.3‰ and an accuracy of < 0.1‰. The values for internal standard are 0.2‰ and 0.3‰, respectively. Replicates yielded a mean precision ($1\sigma$) for the $n$-C$_{29-33}$ alkanes of 0.2‰. Data sets are stored online at PANGAEA (https://doi.pangaea.de/10.1594/PANGAEA.875694).

Statistical analysis has been carried out using the software package PAST 3.0 (Hammer et al., 2001). To calculate correlation

coefficients data sets have been re-sampled at steps of 4 kyr (close to the original average temporal resolution). In order to illustrate the temporal evolution of the dust and $\delta$D$_{wax}$ records, we applied continuous wavelet transform (Morlet) after Torrence and Compo (1998). For this spectral analysis, the Pliocene data sets were linearly interpolated to an even spacing of 5 kyr; the Pleistocene data set to 3 kyr. Significance (p = 0.05) is tested after a chi-squared test with the null hypothesis of a white noise model.





## 3 Results

Plant-wax-derived long-chain $n$-alkane concentrations from ODP Site 659 (Fig. 4) mostly vary between 0.01 0·8 µg·g$^{-1}$ dry sediment and are correlated (Table 1) with the dust record (Tiedemann, 1991). Carbon preference index values mostly exceed 3, indicating terrestrial plant contributions (Eglinton and Hamilton, 1967).

Stable carbon isotope compositions of $n$-alkanes fluctuated between –27.3 and –24.7‰, –26.5 and –24.4‰, and between –26.1 and –23.3‰ for the $n$-$C_{29}$, $n$-$C_{31}$, and $n$-$C_{33}$ alkanes, respectively. A trend to more enriched values started at 3.2 Ma. Isotopic signatures of the major homologues significantly correlate (Table 1). Therefore, we focus on the $n$-$C_{31}$ alkane as the most abundant compound, attributed to the prevailing C4 grass input (Vogts et al., 2009). Spectral analysis (wavelet analysis) indicates that the variability in the isotope record is not stable (Fig. 5). During the periods between 4.7 and 4.65 Ma

and between 3.1 and 3.0 Ma both significant precession and obliquity variability are found in the $\delta^{13}C_{31}$ record. Additional precession variability is found between 3.48 and 3.38 Ma. These periods are characterized by large eccentricity and thus large precession cycle amplitudes. On the other hand, this type of orbital variability breaks down during periods with low precessional variability (4.917-4.775 Ma, 3.520-3.475 Ma, 3.405-3.360 Ma, 3.236-3.190 Ma; see boxes in Fig. 5).

Hydrogen isotope compositions of $n$-alkanes fluctuate between –160 and –124‰, –171 and –133‰, and between –177 and –

127‰ for the $n$-$C_{29}$, $n$-$C_{31}$, and $n$-$C_{33}$ alkanes, respectively. Again, isotopic signatures of the homologues significantly correlate (Table 1) and we thus focus on $\delta D_{wax}$ values of the $n$-$C_{31}$ alkanes, too. Pliocene $\delta D_{31}$ values range from –171‰ to –133‰ (Fig. 4) and show alternating arid and humid conditions in which $\delta D_{31}$ maxima (less negative) indicate aridity corresponding with the dust maxima (Tiedemann, 1991). Spectral analysis indicates some precessional variability around 4.7 Ma, between 3.55 and 3.52 Ma, 3.35 and 3.31 Ma, 3.15 and 3.19, and between 3.04-3.0 Ma (Fig. 5). During low eccentricity

periods little precession variability is found.

A positive correlation is found between $\delta^{13}C_{31}$ and $\delta D_{31}$ for the period between 3.3 and 3.0 Ma, corresponding to the mid-Piacenzian Warm Period (mPWP; 3.264-3.025 Ma after Haywood et al., 2016). This is in contrast to the negative correlation found for the Last Glacial Cycle. The earlier Pliocene results do not reveal a significant correlation between stable carbon and hydrogen isotope compositions of the $n$-$C_{31}$ alkanes (Table 1).

## 4 Discussion

### 4.1 Plant-wax provenance and vegetation sources

The good correspondence between the dust record of ODP Site 659 (Tiedemann, 1991) and the $n$-alkane concentrations indicate predominance of eolian transport of plant waxes probably in the form of coatings on dust particles. Pliocene $\delta^{13}C_{31}$ values display a narrow range around the average of –25.4‰, clearly below the enriched values of up to –23‰ as observed

for the Last Glacial Cycle (Fig. 4; Kuechler et al., 2013). The low $\delta^{13}C_{31}$ variations are attributed to a relatively stable wind system (Tiedemann et al., 1994) and the integration of a large source area. The Pliocene record shows a different pattern than





the Last Glacial Cycle when C3 plant-wax material relatively increased during arid phases, which is attributed to enhanced contributions by the trade winds in combination with no or sparse vegetation cover in the Sahara (Kuechler et al., 2013). This suggests that only small amounts of plant waxes derived from Mediterranean sources because trade winds were weak, in line with the pollen records of ODP Sites 658 (Leroy and Dupont, 1994) and 659 (Vallé et al., 2014). Thus, depleted

$\delta^{13}C_{31}$ values of the Pliocene indicate higher C3 plant coverage at the latitude of the Sahel compared to today suggesting generally more humid conditions. Compared to the Last Glacial Cycle the dominance of C4 plants, estimated up to 90% (Kuechler et al., 2013), is much less prominent during the Pliocene.

Differences in leaf anatomy, rooting depth and photosynthetic pathway may contribute to the final plant-wax $\delta D$ signal (Sachse et al., 2012). Overall, C4 grasses are Deuterium-depleted by ~20‰ relative to C3 trees (McInerney et al., 2011) and

Deuterium-enriched by ~20‰ relative to C3 grasses (Smith and Freeman, 2006). However, the absolute variability of $\delta^{13}C_{31}$ is small (~2‰) and would correspond to a vegetation shift between C3 vs. C4 plants of less than 20%, when using the $\delta^{13}C$ end-members of −35.2‰ for C3 plants and −21.7‰ for C4 plants (Castañeda et al., 2009). The corresponding difference in hydrogen isotopic fractionation would be less than ±4‰. Compared to the large variability in the plant-wax $\delta D$ record (between −171‰ and −133‰ for the $n$-C31 alkane) such plant-dependent variations are minor. Considering the large

uncertainties in the estimation of C4/C3 plant coverage using stable carbon isotopes, we refrain from applying such a correction.

Pollen records from ODP Sites 658 (Leroy and Dupont, 1994) and 659 (Vallé et al., 2014) indicate that Pliocene sub-Saharan savannahs have no modern analog, which harbors the potential for further uncertainties in the plant-wax $\delta D$ record. Nevertheless, $\delta D_{wax}$ studies from offshore of NW Africa covering the African Humid Period (~15-5 kyr) yield a robust

humid signal among different records (Niedermeyer et al., 2010; Collins et al., 2013; Kuechler et al., 2013; Tierney et al. 2017), although the vegetation of this African Humid Period had no modern analog (Watrin et al., 2009). Watrin et al. emphasize that instead of a homogenous latitudinal shift of vegetation zones as a whole, individual plant species likely have an advantage over others. The robustness of the Holocene $\delta D_{wax}$ records implies that this proxy for paleohydrology is apparently not strongly affected by a "non-analogous" vegetation composition.

**4.2 Other effects on the sedimentary $\delta D_{wax}$ signature**

Aridity may exert a considerable influence on the apparent fractionation ($\epsilon$) between plant waxes and meteoric water via evapotranspiration and associated Deuterium-enrichment (Polissar and Freeman, 2010; Douglas et al., 2012; Kahmen et al., 2013a,b). It was found that such an effect is less pronounced in lake sediments compared to soils, likely due to the higher potential of lakes to integrate large catchment areas and the small-scale variability of soils related to differences in

microclimate and vegetation (Douglas et al., 2012). Niedermeyer et al. (2016) could not detect the effect of evapotranspiration comparing high resolution sediment data off West Africa with instrumental records. Also our records of the terrigenous fraction in marine sediments integrate huge catchment areas since large parts of the Saharan and the Sahel can be considered sources of the material of primarily eolian origin (Tiedemann et al., 1994; Vallé et al., 2014). Moreover,





Gao et al. (2014) investigated aerosols from arid and humid subtropical environments and found only minor deviations in ε (< 10‰), attributed to a possible compensation of isotopic enrichment due to decreasing relative humidity by a shift from trees to grasses in the vegetation. Similar results were found in marine surface sediments forming a transect running west along the central African to South African coast (Vogts et al., 2016).

The δD record of the $n$-C$_{31}$ alkane was adjusted for changes in global ice volume (Tierney and DeMenocal, 2013). The Last Glacial Maximum was scaled to 1‰ (Schrag et al., 2002) in the LR04 global benthic $\delta^{18}$O stack (Lisiecki and Raymo, 2005) relative to the present (0‰), and subsequently converted into δD values, using the global meteoric water line. The procedure determined the use of the revised age model (Clemens, 1999), which is based on $\delta^{18}$O variability at ODP Site 659 and thus better matches the LR04 stack (Lisiecki and Raymo, 2005). However, the δD adjustment for the Pliocene period is almost
negligible (<2‰) and results are not shown.

Finally, plant waxes extracted from modern dust samples offshore of West Africa yielded a radiocarbon age of 650±150 years (Eglinton et al., 2002) hinting at residence and transport times of several centuries. Since our research questions concern orbital time scales (and our records have a ~4 kyr resolution), intermediate storage on the continent and mixing of plant waxes are considered to have a negligible effect on our interpretations.

## 4.3 Aridity and humidity in Pliocene West Africa

Correspondence between the δD$_{31}$ and the dust record of ODP Site 659 support previous conclusions concerning the generation of dust during arid periods (Tiedemann et al., 1994) and the use of δD$_{31}$ as humidity proxy. With it we present the first proxy record of West African humidity and precipitation during the Pliocene. This record is also important because river discharge and run-off from northern Africa (including the eastern parts of West Africa) is crucial for the development of
Mediterranean sapropel layers. Almost all recognized sapropels can be correlated with maxima in West African humidity (Fig. 6; δD$_{31}$ minima). In general, our findings highlight the influence of the West African monsoon for Mediterranean sapropel formation, and corroborate a recurrent greening of the Sahara, which might have intermittently allowed hominin migration through otherwise hostile territory (Larrasoaña et al., 2013).

Precipitation in West and northern Africa depends on the West African monsoon mainly varying with low-latitude summer
insolation, which is dominated by 19 kyr and 23 kyr periodicities of the precession cycle (e.g. Kutzbach, 1981, see also Introduction). However, variations in low-latitude insolation cannot explain the entire variability in the δD$_{31}$ and dust records. For instance, four dust percentage maxima between 4.91 and 4.77 Ma correspond to four obliquity minima. Examples of humid periods (strongly depleted δD$_{31}$) during insolation minima were found shortly before 4.90 Ma, centred at 4.79 and 3.50 Ma, or just after 3.20 Ma. Conversely, examples of arid periods (less depleted δD$_{31}$) during insolation maxima
are centred at 4.88, 4.78, 3.48, 337 Ma and just before 3.20 Ma (Fig. 6). Moreover, spectral analysis of both Pliocene (Fig. 5) and Pleistocene (Last Glacial Cycle; Fig. 7) series indicate a more complicated pattern with additional periodicities unrelated to precession. These periods, in which humidity changes do not follow low-latitude insolation, might be linked to obliquity.



Shifts in variability already occurred prior to the intensification of the Northern Hemisphere Glaciation at the beginning of the Pleistocene (2.58 Ma), but are most prominent during the Last Glacial Cycle.

The mechanistic basis for the obliquity signal in low-latitude climate records is a matter of debate. Modelling studies show contrasting results, both including (Tuenter et al., 2003; Weber and Tuenter, 2011) and excluding (Bosmans et al., 2015a; 5 Rachmayani et al., 2016) an influence of the high latitudes. On one hand, obliquity in the tropics is attributed to higher moisture transport from mid/high latitudes (Tuenter et al., 2003) or varying ice sheets and greenhouse gases (Weber and Tuenter, 2011) that influence the monsoon intensity and timing. On the other hand, recent modelling studies revealed an increase in monsoonal rainfall due to a stronger atmospheric pressure contrast between the West African continent (low pressure) and the South Atlantic (high pressure) under conditions of low precession (i.e. maximum local insolation) and high 10 obliquity. Based on this, the interhemispheric (or cross-equatorial) insolation gradient (summer inter-tropical insolation gradient) at lower latitudes was suggested to drive winds and associated moisture transport (Bosmans et al., 2015a,b). In general, insolation gradients shape the climate system by differential heating and thus determine spatial temperature patterns, which ultimately steer atmospheric pressure gradients (Nicholson, 2009).

On orbital time scales, the periodicity of glacial-interglacial cycles with the high-to-low-latitude contrast in insolation (i.e. 15 latitudinal insolation gradient) forces moisture fluxes poleward via strong gradients, thus triggering ice sheet growth (Raymo and Nisancioglu, 2003). In this context, most studies focus on high latitudes and studies considering a latitudinal gradient forcing of tropical monsoon systems are limited to the Last Glacial Cycle (e.g., Davis and Brewer, 2009). However, summer weakening of the latitudinal gradient when obliquity is strong would lead to a northward shift of the tropical monsoon front (Davis and Brewer, 2009). This was recently supported by another climate model, indicating a tight link between the 20 gradient-controlled mid-latitude eddy circulation (driving the heat and moisture transport from low to high latitudes) and the poleward boundary of the tropical rain belt (Mantsis et al., 2014). Thus, a northward shift of the tropical monsoon front is related to increased mid-latitude eddy circulation during summer when the latitudinal temperature gradient linked to the summer latitudinal insolation gradient (dominated by obliquity) is large. For West Africa the strongest monsoonal influence would thus be expected during summer when the latitudinal gradient is weak, shifting the tropical rain belt northward, 25 instead of being purely triggered by local insolation. For the Pliocene, both proxy data and model results confirm reduced meridional and zonal temperature gradients when the tropical monsoon was enhanced (Dowsett et al., 2013; Haywood et al., 2013). Solely low-latitude insolation forcing cannot explain the variations in the $\delta D_{31}$ and dust records.

### 4.3 Towards a more comprehensive understanding of the drivers of the West African Monsoon

Based on our results from the Pliocene and comparison with the Last Glacial Cycle we propose a modified gradient forcing 30 mechanism. Wavelet analyses show repeated shifts in the periodicity of the $\delta D_{31}$ records from the obliquity to the precession band and vice versa, depending on the amplitudes in precession and obliquity (Figs. 5 and 7). We infer a consistent pattern with generally two modes: during periods with strong precession cycles (i.e. high eccentricity), the influence of summer LIG on monsoon variability is superimposed by changes in local insolation. In contrast, during weak precession cycles, especially

those of the Last Glacial Cycle, summer LIG forms the primary forcing of the West African monsoon and its influence on the northward distribution of atmospheric moisture plays the decisive role in the West African climate. This is also supported by climate models, which show a strong dependency of the obliquity-induced precipitation response to precession, but not vice versa (Tuenter et al., 2003). The indication for LIG forcing during weak precession cycles of the Pliocene is less

pronounced than during the Late Pleistocene suggesting that the obliquity forcing of the summer LIG is reinforced by the intensification of the Northern Hemisphere Glaciation. LIG-induced climate shifts during the Last Glacial Cycle are more severe than during the Pliocene due to increased ice-albedo feedbacks (Raymo and Nisancioglu, 2003). In addition, the duration of humid-arid periods seems to increase in the course of the Last Glacial Cycle, reflecting the shift from one dominant forcing to another, i.e. from precession to obliquity. These findings caution against a too simplified view on orbital

forcing mechanisms of the (tropical) hydrologic cycle and hence, related records.

### 5 Conclusion

Our new West African monsoon records suggest that orbital forcing is indeed the major control of monsoon variability, but indicate that changes in local summer insolation are insufficient to explain the full degree of hydrologic variations. We infer a hybrid insolation forcing with two modes, depending on the strength of the precession cycle: during high eccentricity

(large precession fluctuations), the main driver is the precession-controlled local summer insolation, but during low eccentricity, the obliquity-controlled summer latitudinal insolation gradient becomes more important. After the Pliocene, the subsequent intensification of the Northern Hemisphere glaciations would have reinforced the obliquity component in West African monsoon variability via the summer insolation gradient.

**Data availability**

Data sets are stored online at PANGAEA (https://doi.pangaea.de/10.1594/PANGAEA.875694).

**Author contributions**

E. Schefuß designed the research project. Laboratory work was carried out by R.R. Kuechler and E. Schefuß. L.M. Dupont, R.R. Kuechler and E. Schefuß contributed to interpretation and discussion of the results. R.R. Kuechler prepared the manuscript with contributions from all co-authors.

**Competing interests**

The authors declare that they have no conflict of interest.





**Acknowledgements**

We thank B. Beckmann, W. Bevern, T. Evans and M. Yavuz for assistance in the laboratory, S. Clemens for providing the data of the revised age model, and R. Tiedemann for comments and discussion. This study was financially supported by the Deutsche Forschungsgemeinschaft (DFG) through the priority program 527 "IODP/ODP" (Sche903/11) with additional
support by the Research Center/Excellence Cluster "The Ocean in the Earth System" at MARUM – Center for Marine Environmental Sciences, University of Bremen. R.R. Kuechler acknowledges further support by GLOMAR – Bremen International Graduate School for Marine Sciences.

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





**Table 1. Correlation coefficients (r) and (p) for plant-wax-specific stable isotopes (δD, δ¹³C), long-chain *n*-alkane (*n*-C$_x$) and dust percentages at ODP Site 659.**

|  | *n*-Cx | LGC | p | mPWP | p | pre-mPWP | p | EP | p |
|---|---|---|---|---|---|---|---|---|---|
|  | 29/31 | **0.80** | < 0.01 | **0.85** | < 0.01 | **0.70** | < 0.01 | **0.70** | < 0.01 |
| **δD** | 31/33 | **0.74** | < 0.01 | **0.83** | < 0.01 | **0.58** | < 0.01 | **0.63** | < 0.01 |
|  | 29/33 | **0.73** | < 0.01 | **0.79** | < 0.01 | **0.52** | < 0.01 | **0.56** | < 0.01 |
|  | 29/31 | **0.86** | < 0.01 | **0.87** | < 0.01 | **0.82** | < 0.01 | **0.60** | < 0.01 |
| **δ¹³C** | 31/33 | **0.78** | < 0.01 | **0.70** | < 0.01 | **0.64** | < 0.01 | **0.51** | < 0.01 |
|  | 29/33 | **0.62** | < 0.01 | **0.66** | < 0.01 | **0.73** | < 0.01 | **0.41** | < 0.01 |
| **δD vs.** | 29 | *-0.08* | 0.67 | **0.42** | < 0.01 | *0.20* | 0.08 | *0.12* | 0.24 |
| **δ¹³C** | 31 | **-0.55** | < 0.01 | **0.36** | < 0.01 | *-0.04* | 0.76 | *-0.23* | 0.03 |
|  | 33 | **-0.70** | < 0.01 | *0.01* | 0.91 | *-0.10* | 0.38 | *0.05* | 0.61 |
| **δD$_{31}$ vs. dust%** |  | *0.34* | 0.02 | **0.41** | < 0.01 | *0.21* | 0.06 | *0.23* | 0.02 |
| **$n$-C$_{27-35}$ vs. dust%** |  | *0.36* | 0.02 | **0.48** | < 0.01 | *0.16* | 0.16 | **0.70** | < 0.01 |

5  For statistical analyses (PAST 3.0; Hammer et al., 2001), the late Pliocene or Piacenzian interval was split into two periods corresponding to the mid-Piazencian Warm Period (mPWP; 3.273-3.007 Ma) and a preceding one (pre-mPWP; 3.616-3.309 Ma) separated by a stratigraphic gap of ~36 kyr in the sedimentary record. The early Pliocene or Zanclean interval (EP) ranges from 4.997-4.625 Ma. Data sets were re-sampled at evenly spaced 4 kyr, which is close to the original average temporal resolution. Values for the Last Glacial cycle (LGC; last 130 kyr) are given for comparison. Most r-values exhibit p

10 < 0.01 (in bold), less significant values are denoted in italics.





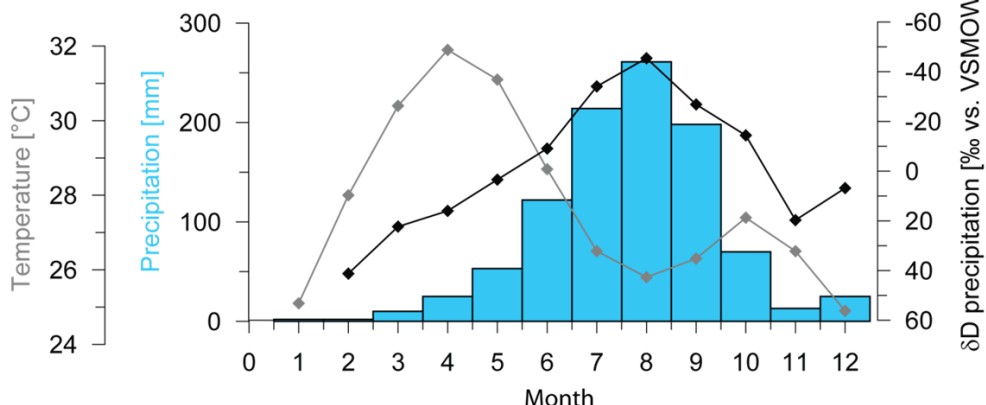

**Figure 1. Monthly precipitation (blue), δD of precipitation (black relative to Vienna Standard Mean Ocean Water) and temperature (grey) at Bamako (Mali, 12°41'24'' N, 7°59'24'' W, 381 m) exemplifying the amount effect (Dansgaard, 1964) in West African hydrology (IAEA/WMO, 2014). This relationship is shown for monthly means, but also holds true on an annual basis**
5 **(Rozanski et al., 1993).**




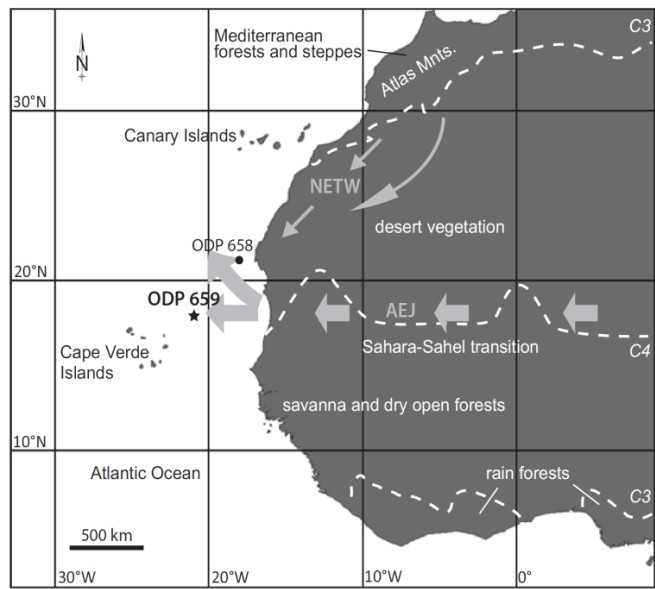

**Figure 2. Location of Ocean Drilling Program (ODP) Sites 659 (star; 18°05'N, 21°02'W, 3070 m water depth) and 658 (dot; 20°45'N, 18°35'W; 2263 m water depth) offshore of West Africa (Ruddiman et al., 1987). Main vegetation zones (dashed lines;**
5   **simplified after White, 1983), dominant photosynthetic pathways (right), atmospheric trajectories (arrows). The thickness of the arrows marks the different altitudes of the African Easterly Jet (AEJ; >3000 m) and the North-East Trade Winds (NETW; <1000 m).**



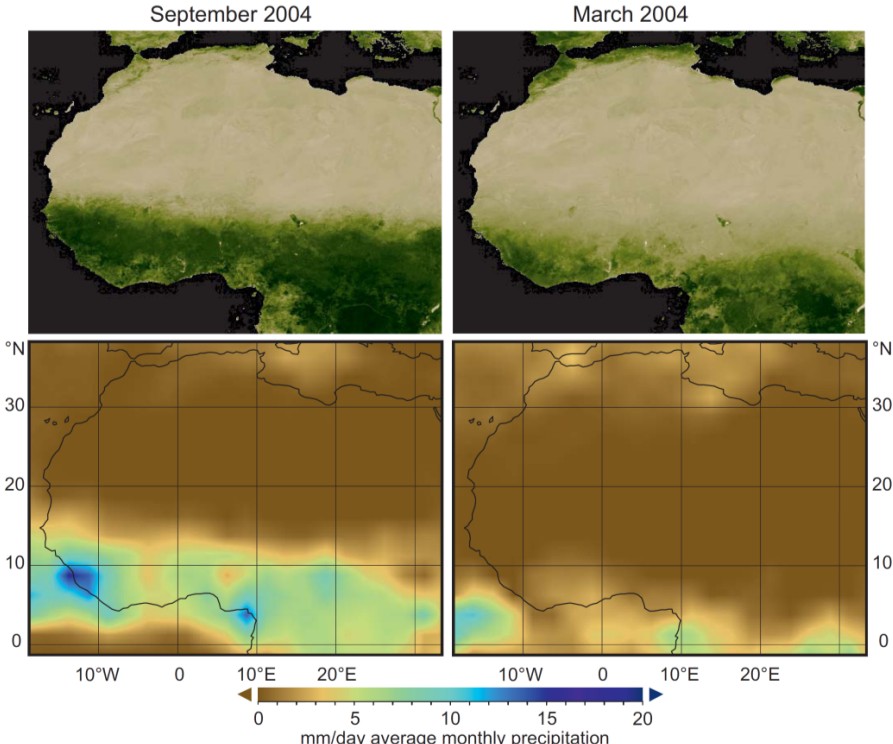

**Figure 3. Seasonal extremes in vegetation (top, NDVI, Normalized Difference Vegetation Index available at www.earthobservatory.nasa.gov) and rainfall (bottom, precipitation in mm per day available at http://www.cdc.noaa.gov/) of northern and West Africa in September 2004 (left) vs. March 2004 (right).**





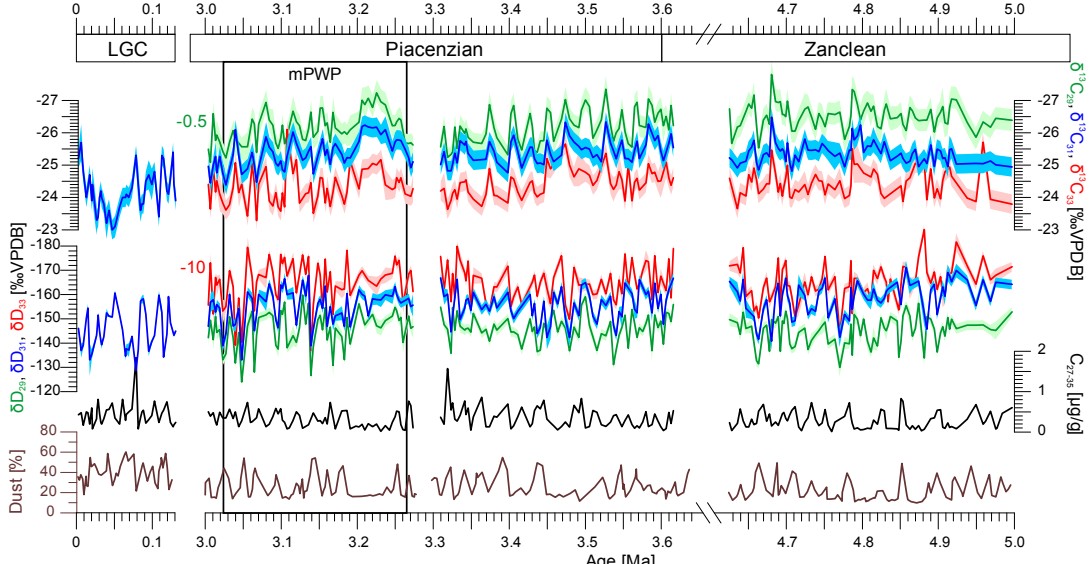

**Figure 4. Results.** From top to bottom: stable carbon ($\delta^{13}$C) and hydrogen ($\delta$D) isotope compositions of $C_{29}$-$C_{33}$ *n*-alkanes in per mil versus Vienna Pee Dee Belemnite (VPDB) and Vienna Standard Mean Ocean Water (VSMOW), respectively; *n*-alkane $C_{27}$-$C_{35}$ concentrations in microgram per gram dry sediment, and dust percentage after Tiedemann (1991) against time in million years (Ma). Age model after Clemens (1999). For visibility, $\delta^{13}C_{29}$ values (green) are shifted minus 0.5‰ and $\delta D_{33}$ values (red) minus 10‰. Results of the $C_{31}$ *n*-alkane for the Last Glacial Cycle (LGC) from Kuechler et al. (2013) are plotted to the left for comparison. mPWP: mid-Piacenzian Warm Period.





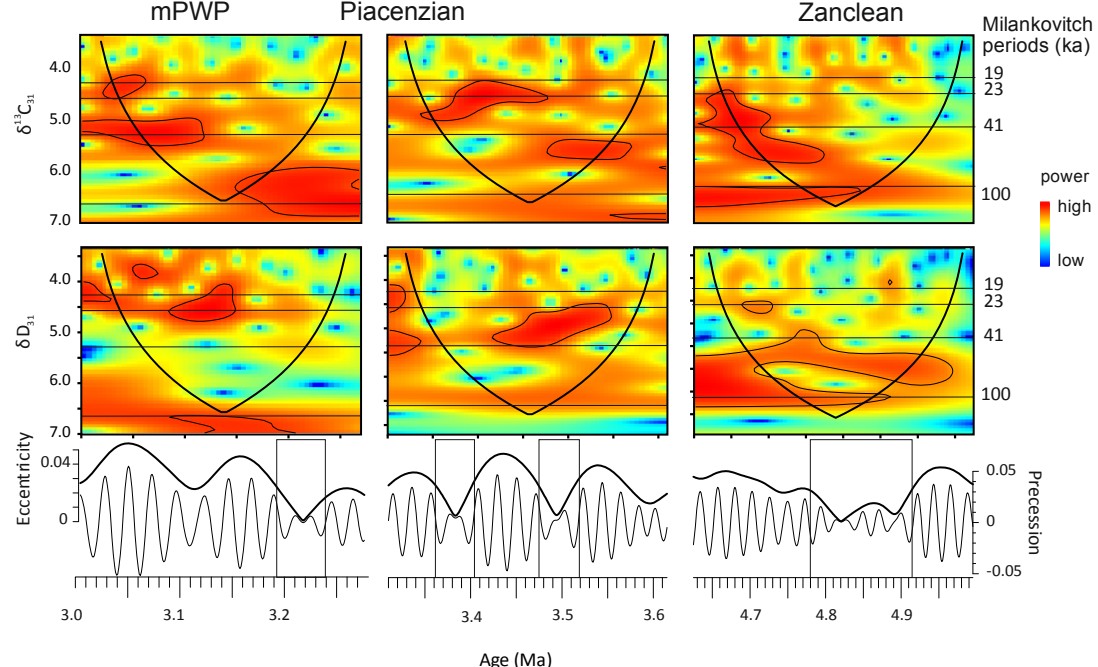

**Figure 5. Spectral analysis of stable carbon ($\delta^{13}C_{31}$) and hydrogen ($\delta D_{31}$) isotope compositions of the $C_{31}$ *n*-alkane using a Mortlet Wavelet for three periods of the Pliocene (4.995-4.625 Ma, 3.615-3.310 Ma, 3.270-3.005 Ma). Data are interpolated to 5 kyr steps. Signal power is expressed in colour shadings with the significance level (p = 0.05) in black contours. The cone of influence (thick black line) identifies the area of boundary effects. Periodicities per thousand years on the vertical axes ($\log_2$-scale). Horizontal lines denote orbital periodicities of 19 and 23 kyr (precession), 41 kyr (obliquity), and 100 kyr. Bottom: eccentricity and precession for the same periods. Boxes indicate periods with reduced eccentricity and low precession amplitudes.**



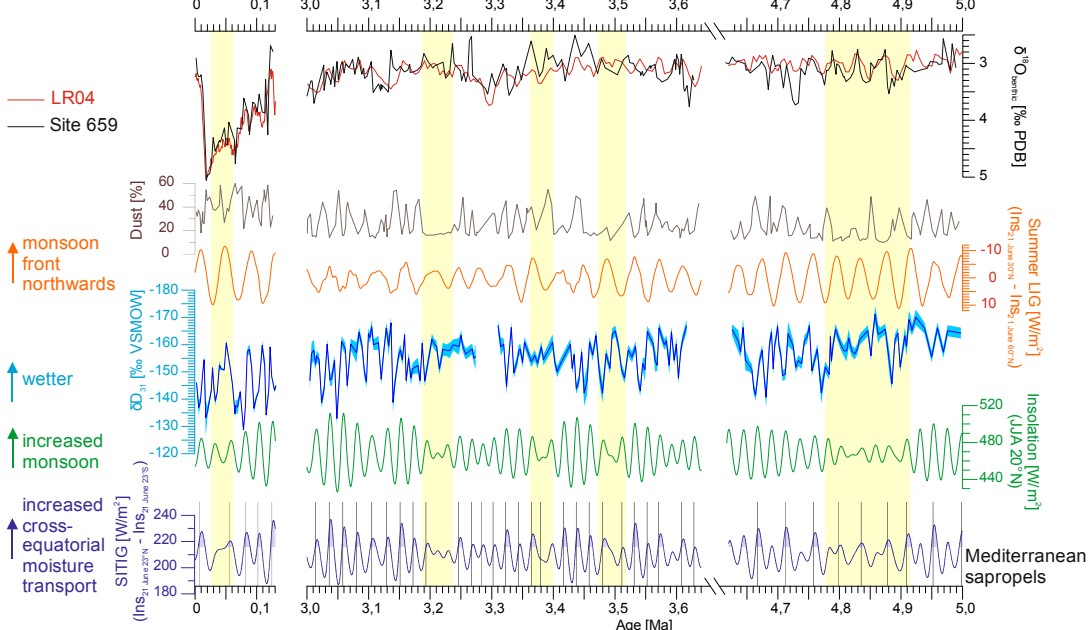

**Figure 6. Multiple isolation forcings. From top to bottom: stable oxygen isotopes of benthic foraminifers ($\delta^{18}O_{benthic}$) from ODP Site 659 (Tiedemann et al., 1994) compared to the global stack LR04 (Lisiecki and Raymo, 2005) in per mil versus Pee Dee Belemnite (PDB); dust percentages after (Tiedemann, 1991); summer latitudinal insolation gradient (LIG) calculated from the 21 June insolation at 30°N minus that of 60°N; Stable hydrogen isotope composition $\delta D_{31}$ in per mil versus Vienna Standard Mean Ocean Water (VSMOW); summer insolation (June, July, August) at 20°N; cross-equatorial summer inter-tropical insolation gradient (SITIG) calculated from the 21 June insolation at 23°N minus that of 23°S and occurrence of Mediterranean sapropels (Emeis et al., 2000). Insolation values from http://vo.imcce.fr/insola/earth/online/earth/online/ (Laskar et al., 2004). Yellow shading denotes periods with low precession variability.**





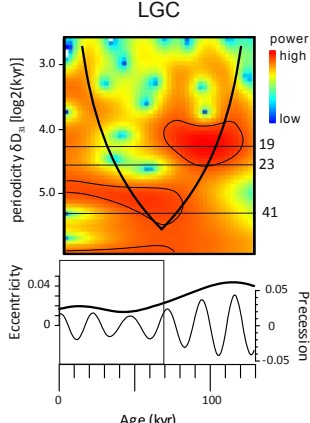

**Figure 7. Spectral analysis of hydrogen ($\delta D_{31}$) isotope compositions of the $C_{31}$ *n*-alkane using a Mortlet Wavelet for the Last Glacial Cycle (LGC). Data are interpolated to 3 kyr steps. Signal power is expressed in colour shadings with the significance level (p = 0.05) in black contours. The cone of influence (thick black line) identifies the area of boundary effects. Periodicities per thousand years on the vertical axes ($\log_2$-scale). Horizontal lines denote orbital periodicities of 19 and 23 kyr (precession), and 41 kyr (obliquity). Bottom: eccentricity and precession for the same period. Box indicate the period with reduced eccentricity and low precession amplitudes.**