# Peer review of "Hybrid insolation forcing of Pliocene monsoon dynamics in West Africa"

_Climate of the Past, 2017_

## Referee Comment (RC1) · Anonymous Referee #1 · 19 Jul 2017

Keuchler, Dupont and Schefuss present new results from marine sediment core off west Africa, which record changes to onshore humidity during the relatively warm Pliocene epoch. The authors have analysed the stable carbon and deuterium/hydrogen isotope compositions of leaf waxes for three intervals of Pliocene climate, with an aim of testing the nature of the hydrological response to insolation forcing during a time window when the additional feedbacks imposed by the growth and retreat of large northern hemisphere ice sheets are not at play. The paper is very significant for its presentation of the first records of humidity changes within the Pliocene from western Africa. The results show different patterns relative to the last glacial cycle, and critically, the authors put forward some interesting new potential mechanisms to link subtropical hydrological changes with insolation forcing. The ideas and data outlined here will stimulate further

research interest in understanding Pliocene circulation changes, but also in determining how and when that system evolved towards the present.

Overall I found the manuscript to be well written and clearly presented. The Introduction is particularly comprehensive and gives both a strong justification for the paper and the relevant literature background to explain the approach. Likewise the opening sections of the Discussion are excellent in assessing some of the potential caveats of the data and its interpretations. The graphics are excellent overall. My main concerns below are minor and are largely for clarification, to ensure that the authors can more clearly demonstrate the strength (or not) of the links between their data and the orbital/insolation forcing.

COMMENTS / CLARIFICATIONS Section 1.1 (page 3) on the regional climatology is good. I feel that it is missing the direct link to the position of the core site (i.e. is ODP 659 in the northern region where there is only one annual peak of precipitation). I assumed that Figure 1 confirmed the single annual peak of precipitation for the core site, but I then realised that this onshore record is located quite far away (∼12*N, 7*W). The position of this field location needs to be on the map of figure 2 to be more explicit in its relationship to the core site, and to confirm that it is also under the same hydrological regime (i.e. not in the zone of the double-annual peak in precipitation). The latter point is particularly important to clarify when the text also notes the role of the Sahara in separating the winter/summer monsoon regimes (line 31).

2. Materials and Methods (1) Although the detailed methods are described elsewhere it would be useful to state here what the composite core depths and/or IODP sample identifiers of the top/bottom samples of the two Pliocene sections are.

(2) Two age models are noted (page 4). How different are these in terms of the timings of events? Would the authors have found the same relationships to insolation if they had used the dust model? I ask because tuning to the LR04 stack in the Pliocene can be challenging given the low signal-to-noise in the original data and in the stack itself,

which could introduce errors in the absolute age and in turn, affect the strength of the conclusions here.

(3) line 28 says that Pleistocene data was used, but this is the first statement of Pleistocene data thus far (the abstract indicates only Pliocene results). Clarify this (perhaps through addressing point (1) above).

Page 5 line 2: strange formatting on the first statement of n-alkane concentrations?

Page 5 line 6: clarify if this is a trend 'down-core' or 'towards present'

Page 5 (Results). The authors discuss the changes to stable isotope variability in relation to precession and obliquity. The spectral analysis results are quite complex and a clear message is not obvious: this likely reflects, in part, the short durations of the records as well as the complex climate relationships that the authors discuss later. This complexity also underpins my earlier comment about how different the two age models for this core site could be: would the same or similar results have been found with the alternative model? Although the caption notes the role of the cone of influence, the text doesn't make clear that statistical significance of orbital periods should not be assigned to e.g. the 100 kyr signal in d13C in the earliest MPWP. The figure 5 caption also needs to clarify that the boxes on the lowermost panel are assigned according to the original orbital parameters and not the data. With these clarifications the strengths and the caveats of the data may be more explicit.

4. Discussion. (1) Page 7 line 20-21: I found it quite difficult to confirm the statement that almost all recognised sapropels could be correlated with the dD31 maxima. I appreciate that Fig 6 is already quite detailed, but perhaps putting asterisks to mark the timing of each sapropel onto the dD31 maxima plot could help here?

(2) The discussion of the links to insolation forcing is good, and acknowledges the difficult relationships between the climate records and the expected insolation forcing (Page 7). Would the generation of phase wheels or some other coherency analysis

help here to visualise the links between the forcings and the feedbacks? It is possible to look at how phasing evolves through time for different proxy records using coherency analysis in evolutive spectra as shown here (phase wheels would have trouble detecting such changes within these narrow windows of time). Such analysis might offer some strength to the arguments made about forcing-response, as well as making figure 6 easier to digest.

(3) When the authors originally discussed the age models they noted that the alternative (Tiedemann) tuned the dust record. It would be interesting to comment in the text about why Tiedemann did this and whether his assumptions have been verified or refuted by this new data (e.g. if he considered only precession to be key, what do the new findings here say about whether that age model could or should be revised?).

---

## Referee Comment (RC2) · Anonymous Referee #2 · 14 Oct 2017

This is an excellent contribution to the understanding of monsoon dynamics in Africa, for the Pliocene time period, on which there is not much information and provides insights on climate dynamics in a warm world. Much of the value of the paper derives from its pretty unique time series of deuterium hydrogen isotopes in long chain n-alkanes derived from higher plant waxes. This is interpreted as a proxy for hydroclimate, and the justification for doing so is well addressed in the paper. To constrain further and strengthen the interpretations of their data, in my opinion the authors should take more into account the uncertainties surrounding the source regions of the n-alkanes in their marine record. The authors do indicate that: "our records of the terrigenous fraction in marine sediments integrate huge catchment areas since large parts of the Saharan and the Sahel can be considered sources of the material of

primarily eolian origin (Tiedemann et al., 1994; Vallé et al., 2014)." "predominance of eolian transport of plant waxes probably in the form of coatings on dust particles" " The low d13C31 variations are attributed to a relatively stable wind system (Tiedemann et al., 1994) and the integration of a large source area"

In my opinion these are important issues that would need to be discussed in more detail in order to clarify, for instance if the biomarker signals are driven by changes in the wind system, source, and/or the modes of transport (e.g. particle sorting with distance, resuspension) of the particulate material rather than just climate at the source.

I would also have provided a more extended discussion on the implications of the difference in magnitude of the isotopic signals interpreted in terms of hydroclimate changes between the late Pleistocene and the Pliocene. Few studies are capable of doing so with the pretty unique data set presented.

The frequency analysis seems quite noisy, and the relevant frequencies at given time intervals are often barely distinguishable from the noise. The discussion would have also benefited from a more detailed assessment of the significance of the frequency analysis in the different time intervals.

---

## Author Comment (AC1) · 24 Oct 2017

COMMENTS / CLARIFICATIONS Section 1.1 (page 3) on the regional climatology is good. I feel that it is missing the direct link to the position of the core site (i.e. is ODP 659 in the northern region where there is only one annual peak of precipitation). I assumed that Figure 1 confirmed the single annual peak of precipitation for the core site, but I then realised that this onshore record is located quite far away (âĹij12°N, 7°W). The position of this field location needs to be on the map of figure 2 to be more explicit in its relationship to the core site, and to confirm that it is also under the same hydrological regime (i.e. not in the zone of the double-annual peak in precipitation).

[Figure]

The latter point is particularly important to clarify when the text also notes the role of the Sahara in separating the winter/summer monsoon regimes (line 31).

Response. We will add the position of the field location to the map in Figure 2. In addition, we'll add the position in Figure 3 with its present-day precipitation and vegetation maps. The main purpose of figure 1 is to illustrate the relation between amount of rainfall and $\delta$D of rain. Its illustration of the modern rainfall regime is of less importance, as we document important differences between the Pliocene and the present.

2. Materials and Methods (1) Although the detailed methods are described elsewhere it would be useful to state here what the composite core depths and/or IODP sample identifiers of the top/bottom samples of the two Pliocene sections are.

Response. We'll add cores and datums to this section. More detailed information can be found at the PANGAEA data base (page 9, line 20).

(2) Two age models are noted (page 4). How different are these in terms of the timings of events? Would the authors have found the same relationships to insolation if they had used the dust model? I ask because tuning to the LR04 stack in the Pliocene can be challenging given the low signal-to-noise in the original data and in the stack itself, which could introduce errors in the absolute age and in turn, affect the strength of the conclusions here.

Response. We acknowledge the critical view on the chosen age model, since we have been aware of this challenge from the beginning of its application in our study. We'll add a small section in the Discussion summarizing the following arguments. First of all, the $\delta$18O record of ODP Site 659 is part of the LR04 stack and thus better matches the LR04 stack, which is necessary for the correction of the $\delta$D record for changes in global ice volume (page 7, lines 5-9). However, there are further reasons for why we decided to apply the $\delta$18O-based age model. In general, both age models (dust and $\delta$18O) contain the same orbital frequencies. However, the dust model assumes only precession as the main forcing and accordingly, Pliocene dust peaks are tuned
to insolation minima. Thus, if using the dust age model, the relationship of $\delta$D to insolation would be the same as for the dust to insolation, which assumes precession forcing. Precession is also important when using the alternative age model based on $\delta$18O, but only for periods with large variations in precession, while obliquity has the stronger impact during times of low precession variability. By application of the dust age model forcing by precession would be exaggerated in the $\delta$D record, since both records (dust and $\delta$D) are well correlated, i.e. arid and humid periods reflected in both records are in accord with each other. Further support for using the $\delta$18O age model provides the comparison of the Pliocene sections with the Last Glacial Cycle, which can be regarded as a reference section for the postulated hybrid insolation forcing. The Last Glacial Cycle has an age model based on $\delta$18O and shows very clearly the two modes of insolation forcing (precession vs. obliquity). These are also apparent during the Pliocene, but less well expressed, attributed to minor ice-albedo feedbacks before the intensification of Northern Hemisphere Glaciation. Having said this, the age models are not very different. We'll prepare a figure that plots the benthic stable oxygen isotopes of ODP Site 659 on both time scales comparing with LR04. We'll provide this figure as supplement (SF 1). It is clear that the adapted time-scale of Clemens (T94R) results in the better correlation with LR04. Additionally, we provide a figure showing the results of REDFIT analyses (Schulz & Mudelsee 2002) for dust % (Tiedemann) and $\delta$Dwax (this study) illustrating that the differences in time scales have very little influence on the position of the significant power maxima (SF 2). A rerun of the wavelet analysis with the original timescale of Tiedemann (T94) also gives very similar results (not shown).

(3) line 28 says that Pleistocene data was used, but this is the first statement of Pleistocene data thus far (the abstract indicates only Pliocene results). Clarify this (perhaps through addressing point (1) above).

Response. The Pleistocene data have been published (Kuechler et al. 2013). They are shown as a comparison. However, this was not mentioned in the caption of table

and figures. We will add the reference. As the paper concerns the Pliocene and not the Pleistocene, the Pleistocene data are not mentioned in the Abstract. To avoid confusion, we'll add in the Abstract that data are compared with those of the Last Glacial Cycle.

Page 5 line 2: strange formatting on the first statement of n-alkane concentrations?

Response. Yes, this error will be corrected into 'between 0.01 and 0.8 $\mu$g g$-1$'.

Page 5 line 6: clarify if this is a trend 'down-core' or 'towards present'

Pesponse. The trend started at 3.2 Ma, which should mean 'towards present'. We'll add the words to the sentence.

Page 5 (Results). The authors discuss the changes to stable isotope variability in relation to precession and obliquity. The spectral analysis results are quite complex and a clear message is not obvious: this likely reflects, in part, the short durations of the records as well as the complex climate relationships that the authors discuss later. This complexity also underpins my earlier comment about how different the two age models for this core site could be: would the same or similar results have been found with the alternative model? Although the caption notes the role of the cone of influence, the text doesn't make clear that statistical significance of orbital periods should not be assigned to e.g. the 100 kyr signal in d13C in the earliest MPWP. The figure 5 caption also needs to clarify that the boxes on the lowermost panel are assigned according to the original orbital parameters and not the data. With these clarifications the strengths and the caveats of the data may be more explicit.

Response. The question on the different outcomes by using different age models has been addressed above in (2). We'll add a reference to Laskar et al. (2004) and the website address to the caption of Figure 6.

4. Discussion. (1) Page 7 line 20-21: I found it quite difficult to confirm the statement that almost all recognised sapropels could be correlated with the dD31 maxima. I

appreciate that Fig 6 is already quite detailed, but perhaps putting asterisks to mark the timing of each sapropel onto the dD31 maxima plot could help here?

Response. We will add an asterisk to each maximum associated with a Mediterranean sapropel. It should be noted that the sapropel record is tuned to Northern Hemisphere insolation maxima at 65 °N (see e.g., Emeis et al., 2000). This implies a better correlation of sapropels and $\delta$D during times of strong precession variability and thus summer insolation, compared to periods of weaker precession variability.

(2) The discussion of the links to insolation forcing is good, and acknowledges the difficult relationships between the climate records and the expected insolation forcing (Page 7). Would the generation of phase wheels or some other coherency analysis help here to visualise the links between the forcings and the feedbacks? It is possible to look at how phasing evolves through time for different proxy records using coherency analysis in evolutive spectra as shown here (phase wheels would have trouble detecting such changes within these narrow windows of time). Such analysis might offer some strength to the arguments made about forcing-response, as well as making figure 6 easier to digest.

Response. The suggested approach of a coherency analysis is ambitious and challenging. This might be part of future work. We feel that our records do not have the resolution nor the length to calculate the evolution of phase relationships. Considering the development of phasing through time, we should take into account that the average sample resolution of the records is ~4 kyr and that detection of leads or lags between the proposed forcing and the climate proxies is limited by the sample resolution and could yield spurious results.

(3) When the authors originally discussed the age models they noted that the alternative (Tiedemann) tuned the dust record. It would be interesting to comment in the text about why Tiedemann did this and whether his assumptions have been verified or refuted by this new data (e.g. if he considered only precession to be key, what do the

new findings here say about whether that age model could or should be revised?).

Response. Tiedemann argued that for the period older between 5 and 2.85 Ma the dust record had the better amplitude to tune. Please remember that at that time there was worldwide only one other high resolution $\delta 18 O$benthic record (ODP Site 846 from the eastern Pacific) to compare with. Nevertheless, years later Clemens managed to tune on the obliquity signal in the stable oxygen record. The Clemens time-scale shows a better fit with the global stack of Lisiecki and Raymo (note that the stack became available another five years later). These are interesting developments in the history of science. However, we feel that it is beyond the scope of our paper. The general assumptions for the Pliocene dust-tuned age model by Tiedemann et al. (1994) are the dominant influence of low-latitude summer insolation (driven by precession), due to the considerably decreased climate variability at high latitudes before the intensification of Northern Hemisphere Glaciations and that during the Pliocene, high-latitude climate variability had little impact on the low latitudes. This might be considered an apriority. Moreover, tuning of the dust record would introduce some circular reasoning in our argument as the dust record partly depends on precipitation. We'll add a small section in the Discussion about the choice of age model listing the arguments mentioned above (under 2 & 3). The implications of the hybrid forcing concept are mentioned in a condensed form at the end of the Discussion section (page 9, lines 9-10): "These findings caution against a too simplified view on orbital forcing mechanisms of the (tropical) hydrologic cycle and hence, related records." These 'related records' are quite a few, including the dust record by Tiedemann, but also the Mediterranean sapropels. We tried to emphasize the general implication of our study by using the word 'caution'. However, with regard to the question on the revision of the dust-based age model, Clemens (1999) already presented a revised age model based on $\delta 18 O$.
* * *
[Figure]

Stable oxygen isotopes of benthic foraminifera of ODP Site 659
on the original time-scale (T94) and
on the adapted one of Clemens (T94R)
compared to LR 04 (Lisiecki & Raymo 2005)

| Kendall's tau correlation | LR 04 | T94 |
|---|---|---|
| T94 | 0,18 | |
| T94R | 0,28 | 0,20 |

**Fig. 1.** SF1

Spectral analysis using REDFIT (Schulz & Mudelsee 2002) comparing the effect of different time-scales –the original dust tuned time-scale T94 in grey (Tiedemann et al. 1994) and the $\delta^{18}O_{benthics}$ based time-scale T94R in black (Clemens 1999)– on the frequency analysis of dust% (Tiedemann 1991) and $\delta D_{31}$ (this study). The anaylsis is carried out in PAST (Hammer et al. 2001) using one segment and "false-alarm lines" based on parametric approximations ($X^2$-test). The Mid-Pliocene part runs from 3.62 to 3.00 Ma and the Early Pliocene part from 5.00 to 4.66 Ma.

[Figure]

**Fig. 2.** SF2

---

## Author Comment (AC2) · 24 Oct 2017

This is an excellent contribution to the understanding of monsoon dynamics in Africa, for the Pliocene time period, on which there is not much information and provides insights on climate dynamics in a warm world. Much of the value of the paper derives from its pretty unique time series of deuterium hydrogen isotopes in long chain n-alkanes derived from higher plant waxes. This is interpreted as a proxy for hydroclimate, and the justification for doing so is well addressed in the paper. To constrain further and strengthen the interpretations of their data, in my opinion the authors should take more into account the uncertainties surrounding the source regions

of the n-alkanes in their marine record. The authors do indicate that: "our records of the terrigenous fraction in marine sediments integrate huge catchment areas since large parts of the Saharan and the Sahel can be considered sources of the material of primarily eolian origin (Tiedemann et al., 1994; Vallé et al., 2014)." "predominance of eolian transport of plant waxes probably in the form of coatings on dust particles" " The low d13C31 variations are attributed to a relatively stable wind system (Tiedemann et al., 1994) and the integration of a large source area" In my opinion these are important issues that would need to be discussed in more detail in order to clarify, for instance if the biomarker signals are driven by changes in the wind system, source, and/or the modes of transport (e.g. particle sorting with distance, resuspension) of the particulate material rather than just climate at the source.

Response. Changes in the wind system affecting the biomarker signal can rather be excluded, based on the comparison with the Last Glacial Cycle (page 6, lines 1-4). Trade winds strengthened not before the beginning of the Pleistocene (Leroy & Dupont, 1994; Vallé et al., 2014) and thus would any influence of the trade winds have been negligible during the Pliocene. Furthermore, residence/transport times of plant waxes extracted from modern dust samples from offshore West Africa yielded a radiocarbon age of 650 $\pm$ 150 years (Eglinton et al., 2002). Since our records have a $\sim$4 kyr resolution, such potential contributions from pre-aged (or re-suspended) plant waxes have a negligible effect on our interpretations (page 7, lines 11-14).

I would also have provided a more extended discussion on the implications of the difference in magnitude of the isotopic signals interpreted in terms of hydroclimate changes between the late Pleistocene and the Pliocene. Few studies are capable of doing so with the pretty unique data set presented.

Response. We'll add the following sentence at the beginning of section 4.3: "In contrast to the stable carbon record, which show much less variability for the Pliocene than in the Last Glacial Cycle, the fluctuations in the Deuterium record are of the same amplitude indicating that large variations between arid and humid periods in West Africa

occurred long before the intensification of Northern Hemisphere Glaciations." We else think that the general discussion on the isotopic signals is sufficient. For instance, the indication of more humid conditions during the Pliocene reflected in the plant-wax $\delta$13C record is mentioned and explained (page 6, lines 1-7). Also with regard to differences in hydroclimate between the Pliocene and the Late Pleistocene, we stress the enhanced influence of the Northern Hemisphere Glaciation on the latitudinal insolation gradient (page 9, lines 4-9).

The frequency analysis seems quite noisy, and the relevant frequencies at given time intervals are often barely distinguishable from the noise. The discussion would have also benefited from a more detailed assessment of the significance of the frequency analysis in the different time intervals.

Response. The significance (against white noise) of the frequency analysis for each time interval is displayed in Figures 5 and 7 (p = 0.05) as is the cone of influence. We'll add a supplementary figure (SF 2) and a supplementary table, which illustrates the significance of the spectral peaks tested against red noise and, in addition, the little impact of the different age models (T94 versus T94R) on the spectral results.

As a response to the questions and comments Referee #1 also raised about our choice of age model we'll add the following short section in the Discussion:

"4.2 Choice of Age model.

We choose the age model based on stable oxygen isotopes advocated by Clemens (1999) over the original one of Tiedemann et al. (1994), since the former is independent from the dust record and produces a better fit with the global benthic $\delta$18O stack (Lisiecki and Raymo, 2005), especially for the Zanclean. In general, both age models (dust and $\delta$18O) contain the same orbital frequencies (supplementary information). However, the dust model assumes precession as the main forcing and accordingly, Pliocene dust peaks are tuned to insolation minima. In case of using the dust age model, the relationship of $\delta$D to insolation would be the same as for the dust to insolation, which means a strong signal associated with precession. This also holds true for the alternative age model based on $\delta$18O, but only for periods with large precession amplitudes, while obliquity has a stronger impact during times of low precession variability. Application of the dust age-model would also exaggerated the precession signal in the $\delta$D record, since both records (dust and $\delta$D) are well correlated. Moreover, tuning of the dust record would introduce some circular reasoning in our argument as the dust record partly depends on precipitation."
* * *
Spectral analysis using REDFIT (Schulz & Mudelsee 2002) comparing the effect of different time-scales –the original dust tuned time-scale T94 in grey (Tiedemann et al. 1994) and the $\delta^{18}O_{benthics}$ based time-scale T94R in black (Clemens 1999)– on the frequency analysis of dust% (Tiedemann 1991) and $\delta D_{31}$ (this study). The anaylsis is carried out in PAST (Hammer et al. 2001) using one segment and "false-alarm lines" based on parametric approximations ($X^2$-test). The Mid-Pliocene part runs from 3.62 to 3.00 Ma and the Early Pliocene part from 5.00 to 4.66 Ma.

[Figure]

**Fig. 1.** SF2

Supplementary Table.

Significant power maxima in periodicity (reciprocal frequency) in ka of dust percentages (Tiedemann 1991) and $\delta D_{31}$ (this study) on the original dust tuned time-scale T94 (Tiedemann et al. 1994) and the $\delta^{18}O_{benthics}$ tuned time-scale T94R (Clemens 1999) used in this study for two windows of the Pliocene: mid-Pliocene from 3.6 to 3.0 Ma and early Pliocene from 5.0 to 4.6 Ma. Significance is based on a $X^2$-test (Hammer et al. 2001). * significant at the 90% level; ** significant at the 95% level; significant at the 99% level.

**Fig. 2.** Supplementary Table Caption

| period | record | age model | T94 | age model | T94R |
|--------|--------|-----------|-----|-----------|------|
| | | 25 | * | 24,4 | ** |
| | dust% | 23 | *** | 23 | *** |
| | | 18 | *** | 18 | *** |
| mid-Pliocene | | 9 | * | 9 | ** |
| | | 125 | ** | 125 | ** |
| | δD$_{31}$ | 25 | ** | 24 | ** |
| | | 10 | *** | 10 | ** |
| | | 40 | ** | 38 | ** |
| | dust% | 24 | *** | 24 | *** |
| | | 18 | * | | |
| early | | 11 | ** | 11 | ** |
| Pliocene | | 55 | * | 53 | ** |
| | δD$_{31}$ | 24 | ** | 23 | ** |
| | | 15 | *** | 16 | *** |
| | | | | 15 | ** |

**Fig. 3.** Supplementary Table

---

## Author Response (AR1)

Dear Luc Beaufort,

Thank you for your kind decision.

All comments made by the anonymous referees are acknowledged and taken into consideration. Comments are uploaded as responses to reviewers on the CP-discussion page and repeated below. Responses are in *italics*, including reference to position in the text. To the revision we add a supplementary of two figures and one table.

With kind regards,
On behalf of Rony Kuechler and Enno Schefuß,

Lydie Dupont

**Response to anonymous Referee #1**

COMMENTS / CLARIFICATIONS Section 1.1 (page 3) on the regional climatology is good. I feel that it is missing the direct link to the position of the core site (i.e. is ODP 659 in the northern region where there is only one annual peak of precipitation). I assumed that Figure 1 confirmed the single annual peak of precipitation for the core site, but I then realised that this onshore record is located quite far away ( 12°N, 7°W). The position of this field location needs to be on the map of figure 2 to be more explicit in its relationship to the core site, and to confirm that it is also under the same hydrological regime (i.e. not in the zone of the double-annual peak in precipitation). The latter point is particularly important to clarify when the text also notes the role of the Sahara in separating the winter/summer monsoon regimes (line 31).

*We will add the position of the field location to the map in Figure 2. In addition, we'll add the position in Figure 3 with its present-day precipitation and vegetation maps. The main purpose of figure 1 is to illustrate the relation between amount of rainfall and δD of rain. Its illustration of the modern rainfall regime is of less importance, as we document important differences between the Pliocene and the present.*

2. Materials and Methods (1) Although the detailed methods are described elsewhere it would be useful to state here what the composite core depths and/or IODP sample identifiers of the top/bottom samples of the two Pliocene sections are.

*We'll add cores and datums to this section. More detailed information can be found at the PANGAEA data base (page 9, line 20).*

(2) Two age models are noted (page 4). How different are these in terms of the timings of events? Would the authors have found the same relationships to insolation if they had used the dust model? I ask because tuning to the LR04 stack in the Pliocene can be challenging given the low signal-to-noise in the original data and in the stack itself,
which could introduce errors in the absolute age and in turn, affect the strength of the conclusions here.

*We acknowledge the critical view on the chosen age model, since we have been aware of this challenge from the beginning of its application in our study. We'll add a small section in the Discussion summarizing the following arguments.*
*First of all, the $\delta^{18}O$ record of ODP Site 659 is part of the LR04 stack and thus better matches the LR04 stack, which is necessary for the correction of the δD record for changes in global ice volume (page 7, lines 5-9). However, there are further reasons for why we decided to apply the $\delta^{18}O$-based age model. In general, both age models (dust and $\delta^{18}O$) contain the same orbital frequencies. However, the dust model assumes only precession as the main forcing and accordingly, Pliocene dust peaks are tuned to insolation minima. Thus, if using the dust age model, the relationship of δD to insolation would be the same as for the dust to insolation, which assumes precession forcing. Precession is also important when using the alternative age model based on $\delta^{18}O$, but only for periods with large variations in precession, while obliquity has the stronger impact during times of low precession variability. By application of the dust age model forcing by precession would be exaggerated in the δD record, since both records (dust and δD) are well correlated, i.e. arid and humid periods reflected in both records are in accord with each other.*

*Further support for using the δ¹⁸O age model provides the comparison of the Pliocene sections with the Last Glacial Cycle, which can be regarded as a reference section for the postulated hybrid insolation forcing. The Last Glacial Cycle has an age model based on δ¹⁸O and shows very clearly the two modes of insolation forcing (precession vs. obliquity). These are also apparent during the Pliocene, but less well expressed, attributed to minor ice-albedo feedbacks before the intensification of Northern Hemisphere Glaciation.*
*Having said this, the age models are not very different. We'll prepare a figure that plots the benthic stable oxygen isotopes of ODP Site 659 on both time scales comparing with LR04. We'll provide this figure as supplement (SF 1). It is clear that the adapted time-scale of Clemens (T94R) results in the better correlation with LR04. Additionally, we provide a figure showing the results of REDFIT analyses (Schulz & Mudelsee 2002) for dust % (Tiedemann) and δD$_{wax}$ (this study) illustrating that the differences in time scales have very little influence on the position of the significant power maxima (SF 2). A rerun of the wavelet analysis with the original timescale of Tiedemann (T94) also gives very similar results (not shown).*

(3) line 28 says that Pleistocene data was used, but this is the first statement of Pleistocene data thus far (the abstract indicates only Pliocene results). Clarify this (perhaps through addressing point (1) above).

*The Pleistocene data have been published (Kuechler et al. 2013). They are shown as a comparison. However, this was not mentioned in the caption of table and figures. We will add the reference. As the paper concerns the Pliocene and not the Pleistocene, the Pleistocene data are not mentioned in the Abstract. To avoid confusion, we add in the Abstract that data are compared with those of the Last Glacial Cycle.*

Page 5 line 2: strange formatting on the first statement of n-alkane concentrations?

*Yes, this error will be corrected into 'between 0.01 and 0.8 µg g⁻¹'.*

Page 5 line 6: clarify if this is a trend 'down-core' or 'towards present'

*The trend started at 3.2 Ma, which should mean 'towards present'. We'll add the words to the sentence.*

Page 5 (Results). The authors discuss the changes to stable isotope variability in relation to precession and obliquity. The spectral analysis results are quite complex and a clear message is not obvious: this likely reflects, in part, the short durations of the records as well as the complex climate relationships that the authors discuss later. This complexity also underpins my earlier comment about how different the two age models for this core site could be: would the same or similar results have been found with the alternative model? Although the caption notes the role of the cone of influence, the text doesn't make clear that statistical significance of orbital periods should not be assigned to e.g. the 100 kyr signal in d13C in the earliest MPWP. The figure 5 caption also needs to clarify that the boxes on the lowermost panel are assigned according to the original orbital parameters and not the data. With these clarifications the strengths and the caveats of the data may be more explicit.

*The question on the different outcomes by using different age models has been addressed above in (2).*
*We'll add a reference to Laskar et al. (2004) and the website address to the caption of Figure 6.*

4. Discussion. (1) Page 7 line 20-21: I found it quite difficult to confirm the statement that almost all recognised sapropels could be correlated with the dD31 maxima. I appreciate that Fig 6 is already quite detailed, but perhaps putting asterisks to mark the timing of each sapropel onto the dD31 maxima plot could help here?

*We will add an asterisk to each maximum associated with a Mediterranean sapropel. It should be noted that the sapropel record is tuned to Northern Hemisphere insolation maxima at 65 °N (see e.g., Emeis et al., 2000). This implies a better correlation of sapropels and δD during times of strong precession variability and thus summer insolation, compared to periods of weaker precession variability.*

(2) The discussion of the links to insolation forcing is good, and acknowledges the difficult relationships between the climate records and the expected insolation forcing (Page 7). Would the generation of phase wheels or some other coherency analysis help here to visualise the links between the forcings and the feedbacks? It is possible to look at how phasing evolves through time for different proxy records using coherency analysis in evolutive spectra as shown here (phase wheels would have trouble detecting such changes within these narrow windows of time). Such analysis might offer some strength to the arguments made about forcing-response, as well as making figure 6 easier to digest.

*The suggested approach of a coherency analysis is ambitious and challenging. This might be part of future work. We feel that our records do not have the resolution nor the length to calculate the evolution of phase relationships. Considering the development of phasing through time, we should take into account that the average sample resolution of the records is ~4 kyr and that detection of leads or lags between the proposed forcing and the climate proxies is limited by the sample resolution and could yield spurious results.*

(3) When the authors originally discussed the age models they noted that the alternative (Tiedemann) tuned the dust record. It would be interesting to comment in the text about why Tiedemann did this and whether his assumptions have been verified or refuted by this new data (e.g. if he considered only precession to be key, what do the new findings here say about whether that age model could or should be revised?).

*Tiedemann argued that for the period older between 5 and 2.85 Ma the dust record had the better amplitude to tune. Please remember that at that time there was worldwide only one other high resolution d$^{18}$O$_{benthic}$ record (ODP Site 846 from the eastern Pacific) to compare with. Nevertheless, years later Clemens managed to tune on the obliquity signal in the stable oxygen record. The Clemens time-scale shows a better fit with the global stack of Lisiecki and Raymo (note that the stack became available another five years later). These are interesting developments in the history of science. However, we feel that it is beyond the scope of our paper.*
*The general assumption for the Pliocene dust-tuned age model by Tiedemann et al. (1994) is the dominant influence of low-latitude summer insolation (driven by*

*precession), due to the considerably decreased climate variability at high latitudes before the intensification of Northern Hemisphere Glaciations and that during the Pliocene, high-latitude climate variability had little impact on the low latitudes. This might be considered an apriority. Moreover, tuning of the dust record would introduce some circular reasoning in our argument as the dust record partly depends on precipitation. We'll add a small section in the Discussion about the choice of age model listing the arguments mentioned above (2 & 3).*

*The implications of the hybrid forcing concept are mentioned in a condensed form at the end of the Discussion section (page 9, lines 9-10): "These findings caution against a too simplified view on orbital forcing mechanisms of the (tropical) hydrologic cycle and hence, related records." These 'related records' are quite a few, including the dust record by Tiedemann, but also the Mediterranean sapropels. We tried to emphasize the general implication of our study by using the word 'caution'. However, with regard to the question on the revision of the dust-based age model, Clemens (1999) already presented a revised age model based on $\delta^{18}O$.*

**Response to anonymous Referee #2**

This is an excellent contribution to the understanding of monsoon dynamics in Africa, for the Pliocene time period, on which there is not much information and provides insights on climate dynamics in a warm world. Much of the value of the paper derives from its pretty unique time series of deuterium hydrogen isotopes in long chain n-alkanes derived from higher plant waxes. This is interpreted as a proxy for hydro-climate, and the justification for doing so is well addressed in the paper.

To constrain further and strengthen the interpretations of their data, in my opinion the authors should take more into account the uncertainties surrounding the source regions of the n-alkanes in their marine record. The authors do indicate that: "our records of the terrigenous fraction in marine sediments integrate huge catchment areas since large parts of the Saharan and the Sahel can be considered sources of the material of primarily eolian origin (Tiedemann et al., 1994; Vallé et al., 2014)." "predominance of eolian transport of plant waxes probably in the form of coatings on dust particles" " The low d13C31 variations are attributed to a relatively stable wind system (Tiedemann et al., 1994) and the integration of a large source area"

In my opinion these are important issues that would need to be discussed in more detail in order to clarify, for instance if the biomarker signals are driven by changes in the wind system, source, and/or the modes of transport (e.g. particle sorting with distance, resuspension) of the particulate material rather than just climate at the source.

*Changes in the wind system affecting the biomarker signal can rather be excluded, based on the comparison with the Last Glacial Cycle (page 6, lines 1-4). Trade winds strengthened not before the beginning of the Pleistocene (Leroy & Dupont, 1994; Vallé et al., 2014) and thus would any influence of the trade winds have been negligible during the Pliocene.*
*Furthermore, residence/transport times of plant waxes extracted from modern dust samples from offshore West Africa yielded a radiocarbon age of 650 ± 150 years (Eglinton et al., 2002). Since our records have a ~4 kyr resolution, such potential contributions from pre-aged (or re-suspended) plant waxes have a negligible effect on our interpretations (page 7, lines 11-14).*

I would also have provided a more extended discussion on the implications of the difference in magnitude of the isotopic signals interpreted in terms of hydroclimate changes between the late Pleistocene and the Pliocene. Few studies are capable of doing so with the pretty unique data set presented.

*We'll add the following sentence at the beginning of section 4.3:*
*"In contrast to the stable carbon record, which show much less variability for the Pliocene than in the Last Glacial Cycle, the fluctuations in the Deuterium record are of the same amplitude indicating that large variations between arid and humid periods in West Africa occurred long before the intensification of Northern Hemisphere Glaciations."*
*We else think that the general discussion on the isotopic signals is sufficient. For instance, the indication of more humid conditions during the Pliocene reflected in the plant-wax $\delta^{13}C$ record is mentioned and explained (page 6, lines 1-7). Also with regard to differences in hydroclimate between the Pliocene and the Late Pleistocene, we stress the enhanced influence of the Northern Hemisphere Glaciation on the latitudinal insolation gradient (page 9, lines 4-9).*

The frequency analysis seems quite noisy, and the relevant frequencies at given time intervals are often barely distinguishable from the noise. The discussion would have also benefited from a more detailed assessment of the significance of the frequency analysis in the different time intervals.

*The significance (against white noise) of the frequency analysis for each time interval is displayed in Figures 5 and 7 (p = 0.05) as is the cone of influence.*
*We'll add a supplementary figure (SF 2), which illustrates the significance of the spectral peaks tested against red noise and in addition the little impact of the different age models (T94 versus T94R) on the spectral results.*
*As a response to the questions and comments that Referee #1 also raised about our choice of age model, we'll add the following short section in the Discussion:*

**"4.2 Choice of Age model**

We choose the age model based on stable oxygen isotopes advocated by Clemens (1999) over the original one of Tiedemann et al. (1994), since the former is independent from the dust record and produces a better fit with the global benthic $\delta^{18}O$ stack (Lisiecki and Raymo, 2005), especially for the Zanclean (Supplementary Figure 1). In general, both age models (dust and $\delta^{18}O$) contain the same orbital frequencies (Supplementary Information). However, the dust age-model assumes precession as the main forcing and accordingly, Pliocene dust peaks are tuned to insolation minima. Using the dust age-model would introduce age-model dependent precession variability. The alternative $\delta^{18}O$-based age-model also has a strong precession signal, but only for periods with large precession amplitudes, while obliquity has a stronger impact during times of low precession variability. Moreover, tuning of the dust record would introduce some circular reasoning in our argument as the dust record partly depends on precipitation."